# Effects of Cellulase and Xylanase on Fermentation Characteristics, Chemical Composition and Bacterial Community of the Mixed Silage of King Grass and Rice Straw

**DOI:** 10.3390/microorganisms12030561

**Published:** 2024-03-12

**Authors:** Chenchen Qiu, Nanbing Liu, Xiaogao Diao, Liwen He, Hanlin Zhou, Wei Zhang

**Affiliations:** 1State Key Laboratory of Animal Nutrition, College of Animal Science and Technology, China Agricultural University, Beijing 100193, China; qccydsa@163.com (C.Q.); b20233040437@cau.edu.cn (N.L.); helw@cau.edu.cn (L.H.); 2Sanya Institute of China Agricultural University, Sanya 572024, China; b20193040335@cau.edu.cn; 3Key Laboratory of Ministry of Agriculture and Rural Affairs for Germplasm Resources Conservation and Utilization of Cassava, Key Laboratory of Ministry of Agriculture and Rural Affairs for Crop Gene Resources and Germplasm Enhancement in Southern China, Tropical Crops Genetic Resources Institute, Chinese Academy of Tropical Agricultural Sciences, Danzhou 571737, China; 4Zhanjiang Experimental Station, Chinese Academy of Tropical Agricultural Sciences, Zhanjiang 524000, China

**Keywords:** fibrolytic enzyme, king grass, mixed silage, rice straw, silage quality

## Abstract

This study was to investigate the effects of cellulase and xylanase on fermentation characteristics, nutrient composition and the bacterial community of the mixed silage of king grass and rice straw. Lab-scale bag silage was produced and seven groups were studied: blank control (CK); added 1%, 2% cellulase (CE1, CE2); added 1%, 2% xylanase (XY1, XY2); and added 0.5% cellulase +0.5% xylanase, 1% cellulase +1% xylanase (CX1, CX2). The results showed that the application of additives in six treated groups exerted a positive effect on lactic acid (LA) content and their pH values decreased significantly (*p* < 0.05). The addition of cellulase and xylanase decreased (*p* < 0.05) the content of neutral detergent fiber (NDF) and acid detergent fiber (ADF) significantly and increased (*p* < 0.01) the crude protein (CP) and water-soluble carbohydrate (WSC) content. Filter paper enzyme activity (FPA) declined and xylanase activity (XA) intensified (*p* < 0.05) as ensiling was prolonged, where most of the enzymatic treatments (especially XY2, CX2) resulted in increased enzyme activities. Moreover, the addition of cellulase and xylanase reduced the abundance of harmful bacteria such as *Acinetobacter* and *Klebsiella* and increased the abundance of lactic acid bacteria such as *Lacticaseseibacillus*, *Lactiplantibacillus*. In conclusion, the addition of cellulase and xylanase would improve fermentation quality and nutrient preservation via altering the bacterial community, with 1% cellulase or complex enzyme best.

## 1. Introduction

King grass belongs to the *Pennisetum* genus of the *Poaceae* or *Gramineae* family, which exhibits a wide distribution in subtropical and tropical regions across the globe [1]. The growth of king grass exhibits a robust pattern during the summer rainy season but experiences a decline in yield by winter. Therefore, it is crucial to address the issue of preserving king grass in order to ensure a sufficient supply for ruminants during the winter period [2]. Ensiling is a preservation technology that allows the anaerobic fermentation of green fodder under sealed conditions in order to maintain its quality in a relatively stable manner over a longer period of time [3]. It is a prevailing way to preserve and supply year-round availability of moist feedstock. In essence, ensiling is a process of competition between lactic acid bacteria and spoilage bacteria, which is dictated by substrate characteristics, environmental temperature, silage additives, etc. For grasses in the *Poaceae* family, the most suitable moisture content for silage should be between 65% and 75%. Excessive moisture content is not conducive to the growth of lactic acid bacteria and can also cause nutrient loss due to the compression of raw material. However, high moisture (above 80%) and fiber contents, low water-soluble carbohydrate (WSC) content and poor epiphytic lactic acid bacteria flora of fresh king grass make it difficult to ensile directly, likely leading to severe nutrient loss and poor fermentation quality [4].

Moisture content is a dictating factor for successful silage production. Due to its exceptional water adsorption properties, straw is frequently utilized as a moisture absorber in practical applications [5]. Rice straw is one of the most common byproducts that is under exploited; only about 20% is used for industrial production and animal feed. There are numerous studies demonstrating that the incorporation of rice straw into high moisture silage can effectively mitigate nutrient loss and enhance fermentation quality [6,7,8]. Thus, it is hypothesized that mixing rice straw could help improve the fermentation quality of king grass silage. However, rice straw is also low in WSC content with high lignocellulose content, so low digestibility and poor feeding value might be an obstacle of the mixed silage.

Conventional silage feed raw materials should meet the following conditions: they should contain an appropriate amount of fermentation substrate and water-soluble carbohydrates, have a dry matter content of more than 200 g/kg, low buffering capacity, an ideal physical structure, and be easy to compact in a suitable silage container [3]. If the raw material does not meet the above conditions, additives are needed to improve the fermentation quality of silage. Cellulase and xylanase are complex enzymes that can, respectively, decompose cellulose and xylan into monosaccharides. They are frequently employed as additives in silage to facilitate the breakdown of the intricate cell wall composition of plants, providing substrates for the fermentation process carried out by lactic acid bacteria during ensiling [9]. Several studies have shown that the addition of cellulase and xylanase can increase the number of lactic acid bacteria, reduce the abundance of harmful microorganisms and improve the quality of silage fermentation [10,11]. When cellulase and xylanase are used as feed additives, they can also improve the growth performance of ruminants by altering the rumen microflora [12]. Thus, it is believed that the application of these enzymes would promote the fermentation of mixed silage of king grass and rice straw, finally improving its feeding value. So far, there is a lack of research regarding the utilization of cellulase and xylanase enzymes in the mixed silage of king grass and rice straw, and the degradation pattern of fiber in this mixed silage remains uncertain.

In the present study, fresh king grass was mixed with a given amount of air-dried rice straw to achieve an ideal moisture content and then ensiled with the addition of cellulase and xylanase at various doses (0, 1.0%, 2.0%), mainly focusing on fermentation parameters, nutrient composition, and the bacterial community of mixed silage of king grass and rice straw.

## 2. Materials and Methods

### 2.1. Silage Preparation

King grass was harvested on 29 May 2022 at the affiliated experimental base of the Chinese Academy of Tropical Agriculture (109°50′ E, 19°51′ N), with a harvest height of about 2 m. Fresh king grass was cut to 1–2 cm with a grass shredder (Donghong No.1, Donghong Mechanical Equipment Co., Ltd., Beijing, China) and then was mixed with prepared air-dried rice straw (about 2 cm) at a ratio of 9:1. The mixture was ensiled with the addition of different fibrolytic enzymes on a fresh matter basis, i.e., (1) blank control (CK), (2) 1% cellulase (CE1), (3) 2% cellulase (CE2), (4) 1% xylanase (XY1), (5) 2% xylanase (XY2), (6) 0.5% cellulase +0.5% xylanase (CX1), (7) 1% cellulase +1% xylanase (CX2). After being thoroughly mixed, the materials were ensiled in lab-level polyethylene silage bags (500 g per bag) and were vacuum-sealed using a domestic vacuum machine. A total of 63 bags (7 treatments × 3 ensiling durations × 3 replicates) were individually prepared and stored at natural temperature (26–30 °C). Three bags of each treatment were randomly unsealed for sample collection on day 3, 7 and 30 of ensiling, to determine fermentation quality, chemical composition, fibrolytic enzyme activity and bacterial community. Additionally, cellulase (enzyme activity ≥ 5 × 10^4^ U/g) was obtained from Shandong Longkete Enzyme Preparation Co., Ltd. Xylanase (Yishui, China) (enzyme activity ≥ 5 × 10^4^ U/g) was obtained from Henan Yangshao Biochemical Engineering Co., Ltd. (Yangshao, China).

### 2.2. Chemical Composition, Fermentation Parameter and Enzyme Activity Analysis

Samples were heated at 65 °C for 72 h to determine the dry matter (DM) content, and ground (1 mm sieve) for chemical composition analysis. Crude protein (CP) was measured using the Kjeldahl method following the procedure of the Association of Official Analytical Chemists [13]. Neutral detergent fiber (NDF), acid detergent fiber (ADF) and acid detergent lignin (ADL) were determined by Van Soest et al. [14]. Water-soluble carbohydrate (WSC) was determined according to a previously described method [15]. Extraction fluid was prepared by immersing 10 g of fresh silage with 90 mL distilled water at 4 °C for 24 h. The filtrate was used to measure pH with a portable pH meter, and its ammonia-N content was determined according to the method of Ke [16]. The organic acids (lactic acid, acetic acid, propionic acid and butyric acid) were analyzed using high-performance liquid chromatography [17].

Filter paper enzyme activity (FPA) and xylanase activity (XA) were measured using the extraction fluid as described above. FPA and XA were determined using the methods of Dunn et al. and Nordmark et al. [18,19].

### 2.3. Bacterial Community Analysis

The silage samples collected on day 3, 7, and 30 for the treatment groups of 1% enzyme level and the control group (CK) were dedicated to bacterial community analysis by 16S rDNA sequencing technology. Briefly, DNA was extracted using the TGuide S96 Magnetic Soil/Stool DNA Kit (Tiangen Biotech (Beijing, China) Co., Ltd.) according to manufacturer instructions. A universal primer (338F: 5′-ACTCCTACGGGAGGCAGCA-3′, 806R: 5′-GGACTACHVGGGTWTCTAAT-3′) was used to amplify the V3-V4 region of 16S rRNA gene from the genomic DNA. Following purification and quantification, the amplicons were sequenced on Illumina novaseq 6000 (Illumina, Santiago, CA, USA). After sequencing, raw data were primarily filtered by Trimmomatic (version 0.33). Identification and removal of primer sequences were processed by Cutadapt (version 1.9.1). PE reads were assembled by USEARCH (version 10), followed by chimera removal using UCHIME (version 8.1), and the high-quality reads were used in the following analyses. Sequences with a similarity of ≥97% were clustered into the same operational taxonomic unit (OTU) by USEARCH (version 10.0), and the OTUs with reabundance <0.005% were filtered. According to the OTU results, alpha (Shannon, Simpson, Chao1, Ace and coverage) and beta diversity (principal coordinate analysis [PCoA]) were obtained using QIIME (Version 2.15.3) and R software (R Foundation for Statistical Computing, Vienna, Austria), respectively.

### 2.4. Statistical Analysis

The collected data were analyzed using the general linear model procedure in IBM SPSS Statistics for Windows, version 26.0 (IBM, Armonk, NY, USA). Duncan’s test was used for multiple comparisons and significant differences were declared when *p* < 0.05.

## 3. Results

### 3.1. Chemical Composition of the Raw Materials for Silage Production

Dry matter (DM), crude protein (CP), neutral detergent fiber (NDF), acid detergent fiber (ADF), and acid detergent lignin (ADL) contents of fresh king grass and rice straw are summarized in Table 1. Compared to rice straw, king grass has a lower dry matter content and higher crude protein content. The levels of neutral detergent fiber, acidic detergent fiber, and acidic detergent lignin in the two are not significantly different.

### 3.2. Fermentation Quality of Mixed Silage of King Grass and Rice Straw

The dynamic changes in pH, organic acid content and NH_3_-N content of the mixed silage of king grass and rice straw are shown in Table 2. In the present study, the NH_3_-N content of the mixed silage increased as ensiling duration prolonged, where the values on day 30 of ensiling were higher (*p* < 0.01) than those on day 3 or day 7 of ensiling. As for additive treatments, the addition of cellulase or xylanase resulted in a decrease (*p* < 0.01) in silage pH and NH_3_-N content as well as an increase (*p* < 0.01) in lactic acid (LA) and acetic acid (AA) concentrations, but the dose effect of enzymes was not remarkable. In detail, the pH values of all enzyme-added groups (CE1, CE2, XY1, XY2, CX1, CX2) were lower (*p* < 0.01) than those of the CK group on each sampling day, and no difference was found between the groups of different enzymes or doses. LA concentration was numerically higher in all the enzyme-added silages relative to those in the CK group on each sampling day, where the LA concentrations in CE2 and XY2 were significantly higher (*p* < 0.01) than those in CK group by day 3 of ensiling, and that of XY1 was also higher (*p* < 0.01) by day 7 of ensiling. The AA contents of CK and CX2 at day 30 of ensiling were significantly higher (*p* < 0.05) than those at day 3 and 7 of ensiling. The NH_3_-N contents of the CE2, XY1, XY2, CX1 and CX2 groups were lower (*p* < 0.01) than those of the CK group on each sampling day except for those of CX1 on day 3 and day 7, and that of XY2 on day 7. In addition, propionic acid (PA) and butyric acid (BA) were not detected in all the silages in the present study.

### 3.3. Chemical Composition of Mixed Silage of King Grass and Rice Straw

The dynamics of DM, WSC, NDF, ADF and ADL content of the mixed silage of king grass and rice straw are presented in Table 3. In the present study, ensiling time and additives exerted significant effects (*p* < 0.05) on the content of WSC, NDF, ADF, and ADL in the mixed silages but not on DM content (*p* < 0.05), along with significant interaction effects (*p* < 0.05) on WSC and ADF contents. Specifically, the WSC content of all treatment groups increased and NDF content decreased as ensiling fermentation prolonged, where WSC values were much higher (*p* < 0.01) and NDF concentration was lower on day 30 of ensiling than those on day 3 or day 7. The WSC contents of the CE2, XY1, XY2, CX1 and CX2 groups were higher (*p* < 0.01) than those of the CK group on day 7 and day 30 of ensiling, along with those of CE1 and CE2 which were higher on day 3 of ensiling. The NDF and ADF contents of each additive treatment group were lower (*p* < 0.05) than those of the CK group at each timepoint. Furthermore, inconsistent changes caused by ensiling time or additive were found in ADL content (*p* < 0.05).

### 3.4. Enzyme Activity of Mixed Silage of King Grass and Rice Straw

The dynamics of filter paper enzyme activity (FPA) and xylanase activity (XA) in the mixed silage of king grass and rice straw are shown in Figure 1. On the 3rd day of ensiling, FPA was higher in the enzyme treatment groups, with the highest enzyme activity of 134.70 U/mL in XY2 group. With the extension of ensiling time, the FPA of the control group CK declined to 0, and that of group XY2 and CX2 decreased significantly (*p* < 0.01). The XA was detected in all treatment groups on the 3rd day of ensiling, with the highest value in the CK. The activity of XA increased in all treatment groups on the 30th day of ensiling, and that of groups XY2, CX1 and CX2 significantly increased (*p* < 0.05) relative to their corresponding values on the 3rd day.

### 3.5. Bacterial Community of Mixed Silage of King Grass and Rice Straw

The alpha diversity of the bacterial community in the mixed silage of 1% additive groups and the CK group is shown in Table 4. In the present study, the values of sequencing coverage were over 0.99 for all the silage samples. In general, ensiling time and additives exerted significant effects (*p* < 0.01) on the Shannon and Simpson index along with significant interaction (*p* < 0.01), where their values on day 30 of ensiling were remarkably higher in the enzyme-added groups relative to those in the CK group. Moreover, the values of the XY1 and CX1 groups were much higher (*p* < 0.01) on day 30 of ensiling relative to those on day 3 or day 7 of ensiling. Furthermore, the Chao1 index in all groups increased as ensiling duration prolonged, where the values on day 30 of ensiling were higher (*p* < 0.01) than those on day 3 or day 7 of ensiling. Similarly, The Ace index on day 7 and 30 was significantly higher (*p* < 0.01) than on day 3 in the CK group. However, no difference in Chao1 and Ace index was found in the treatments (*p* > 0.05).

Principal coordinate analysis (PCoA) by the weighted uniFrac method (Figure 2) illustrated that the samples of the additive groups were cross-distributed with those of the CK group on day 3 and day 7 of ensiling but were clustered separately from those of the CK group on day 30 of ensiling as well as their corresponding samples on day 3 and day 7.

The relative abundance of the bacterial community on phylum level is illustrated in Figure 3. Firmicutes (39.51–53.61%) and Proteobacteria (33.86–49.03%) were the two dominant phyla in the mixed silage of king grass and rice straw. In comparison, the relative abundance of phyla Firmicutes and Cyanobacteria was not significantly affected by ensiling time (*p* > 0.05). The relative abundance of Proteobacteria in CE1 decreased (*p* < 0.05) whereas the relative abundance of Bacteroidota in CX1 increased (*p* < 0.05) as ensiling fermentation was extended.

The abundance on a genus level is shown in Figure 4. *Acinetobacter* (14.32–31.23%) was the dominant genus across the ensiling process, whereas silage LAB like *Lactiplantibacillus* (5.05–15.25%), *Lacticaseibacillus* (1.78–14.56%), *Companilactobacillus* (4.02–9.77%), *Lactococcus* (1.15–8.04%), *Weissella* (1.79–4.98%), *Limosilactobacillus* (0.41–8.48%) and *Levilactobacillus* (0.55–5.20%) had a low abundance. The relative abundance of *Acinetobacter*, *Klebsiella* in CE1, XY1 and CX1 decreased as ensiling fermentation went on. On day 3 of ensiling fermentation, *Lacticaseibacilli* was the dominant genus of LAB. After a 30-day fermentation period, *Lactipalantibacillus* gradually became the dominant LAB genus, and the abundance of *Lactiplantisbacilli* in the CK group was higher than that in other groups (*p* < 0.05). Additionally, the bacterial genera with relative abundance over 1% accounted for a less gross abundance in the treatment groups than that in the CK group, i.e., more genera were grouped into the section “others”.

## 4. Discussion

pH value is an indicator that intuitively reflects the fermentation quality of silage, and LA is an important reason for the decrease in silage pH [20]. In the present study, the addition of fibrolytic enzymes resulted in the decrease in pH value of the mixed silage and the increase in LA content. This is due to the fact that cellulase and xylanase could destroy the complex cell wall structure of plants, decompose the structural polysaccharides into monosaccharides or small molecular sugars that can be used by lactic acid bacteria for fermentation, whereby producing a large amount of LA to reduce the pH of silage [21,22]. Similarly, the study of Zhao et al. also showed that cellulase could promote LA production and reduce the pH of silage [23]. AA is produced by heterofermentative lactic acid bacteria [24]. The AA content of each treatment group gradually increased as ensiling fermentation was extended, but there was no significant difference overall. This indicates that cellulase and xylanase had less effect on heterogeneous fermentation in silage.

Generally, NH_3_-N reflects the breakdown of amino acids and proteins in the silage. A higher content of NH_3_-N in the silage indicates that more proteins and amino acids are broken down in the feed, and the quality of the silage will be inferior [25]. In this study, though the NH_3_-N contents of all treatment groups increased significantly by day 30 of ensiling, those of the enzyme treatment groups were lower than those of CK, which might be due to the fact that cellulase and xylanase could indirectly provide sufficient substrate for lactic acid bacteria fermentation and produce LA, thus reducing the pH value, and the lower pH condition could inhibit the activity of plant protease and reduce protein decomposition [26]. In addition, PA and BA were not detected in each treatment group in this experiment, possibly because the pH dropped to a low level at the early stage of silage, which inhibited the growth of PA producing *Propionibacterium*, *Clostridium propionate* and BA producing *Clostridium butyricum* [27].

DM content is positively correlated with the nutrient content of silage, and the respiration of plant cells and microbial fermentation will cause the loss of DM during the ensiling process [28]. In the present study, there was no significant change in DM content in all treatment groups with the prolongation of ensiling time. It is indicated that enzyme additives have less effect on the DM content of the mixed silage. WSC is an important factor in determining the success of silage preparation, as a substrate for lactic acid bacteria fermentation and can improve the carbon source for lactic acid bacteria proliferation [29]. In this study, the WSC content of all treatment groups was higher than that of CK, which was due to the fact that cellulase and xylanase could promote the degradation of cell walls of king grass and rice straw to soluble sugars. In addition, with the extension of ensiling time, NDF and ADF content in the mixed silage gradually decreased, and the NDF content was lower in the mixed group of cellulase and xylanase. It might be because xylanase has the ability to degrade hemicellulose, which can increase the specific surface area and average pore size of plant cellulose and increase the contact area between enzymes and cellulose, and the synergistic effect of two enzymes is beneficial to the degradation of the fibrous material of king grass and rice straw [30]. ADL is the main barrier affecting the degradation of structural polysaccharides such as cellulose and hemicellulose, and the higher the ADL content, the more unfavorable the enzymatic digestion [31]. In the present study, the ADL content of the enzyme treatment group was not significantly different from CK.

Cellulase is a group of complex enzymes that can decompose cellulose into monosaccharides, including endo-glucanases, exo-glucanases and β-glucosidases. This enzyme family can synergistically promote plant cell wall degradation [32]. Filter paper enzyme activity is the result of the synergistic action of endoglucanase, exo-glucanase and β-glucosidase, which represents the total effect of cellulose degradation, and its activity represents the total cellulase activity in the silage process [33]. The magnitude of filter paper enzyme activity (FPA) in silage is not only influenced by temperature, pH and fermentation substrate but is also related to the cellulolytic enzyme producing bacteria. Lee et al. demonstrated that *Weissella* spp. of the Firmicutes are able to produce β-glucosidase and participate in fiber degradation [34], whereas most *Weissella* spp. predominate mainly in the early stage of ensiling [35]. This also corroborates the result that the FPA could be detected in the early stage of ensiling but not in the later stage in the CK of this study. Cellulase activity is not only related to bacteria, but also has a certain relationship with cellulose content [36]. In this study, the enzyme treatment group had a higher enzyme activity of filter paper in the early stage of ensiling with a declining overall trend. This must be related to the enzyme activity attached to the additive itself, and the fiber content in the enzyme treatment groups gradually decreased as ensiling fermentation went on. Its loose organizational structure may favor the attachment of more microorganisms and the subsequent production of cellulase [37]. Xylanase is a group of enzymes capable of decomposing xylan. Because xylan is the main hemicellulose component in plants, xylanase activity (XA) reflects the degradation of hemicellulose in plant tissues. Williams and Orpin have reported that xylanase can be synthesized by some microorganisms and some substrates with complex structures have a strong induction effect on xylanase [38]. As structurally complex substrates, king grass and rice straw are beneficial for inducing more xylanase release in some bacteria. The lignin content may affect the attachment and utilization of microorganisms, and higher lignin content is not conducive to the attachment of microorganisms. This may be related to the increase in XA in the late stage of ensiling compared to the early stage of ensiling in each treatment group in this study.

Silage fermentation is a process in which attached LAB ferment substrates to produce LA and reduce pH, promoting LAB proliferation to become dominant bacteria while inhibiting the growth of harmful bacteria [39]. Studying the microbial diversity and composition structure in silage would help figure out the effect of treatments. For all samples, average coverage was over 0.99, making it reliable to analyze the microbial community. This study showed that the addition of cellulase and xylanase increased the Shannon and Simpson indexes of the bacterial community in the mixed silage relative to the CK group, indicating an increase in microbial diversity and richness. This result was similar to that of Zhao et al. [23]. The Chao1 and Ace indexes of the bacterial community in all the silage groups gradually increased as the ensiling time prolonged, indicating that the ensilage process increased the richness of bacterial species.

On the phylum level, Firmicutes and Proteobacteria were the dominant phyla in the mixed silage during the ensiling process. Firmicutes are acid hydrolytic microorganisms that can proliferate under low pH conditions [40]. Gram-negative bacteria such as Escherichia coli included in Proteobacteria are generally considered as harmful bacteria for silage fermentation [41]. In the present study, the abundance of Firmicutes gradually increased and the abundance of Proteobacteria gradually decreased in the cellulase group with the extension of ensiling time, which indicates that cellulase could promote the growth of Firmicutes in silage and inhibit the proliferation of undesirable fermenting bacteria. Consistently, Mu et al. found that the most important phyla in the mixed silage of high-moisture amaranth and rice straw were Firmicutes and Proteobacteria, and the addition of cellulase could increase the relative abundance of Firmicutes [42].

On the genus level, *Acinetobacter* was the dominant genus during the ensiling process, and *Klebsiella*, *Lactiplantibacillus*, *Lacticaseseibacillus*, *Companilatobacillus*, *Lactococcus*, *Weissella* were also detected in the mixed silage. *Acinetobacter* is an aerobic bacterium, which should not appear in silage under normal circumstances [43]. Research has shown that some *Acinetobacter* can survive when acetic acid is sufficiently present in the surrounding environment [44]. In this study, considering the presence of *Acinetobacter* in the mixed silage and the increase in acetic acid levels with the prolongation of ensiling time, it is speculated that some *Acinetobacter* utilize acetic acid to survive in the anaerobic stage of silage. Chen et al. also detected *Acinetobacter* in the king grass silage [45]. *Klebsiella*, a facultative anaerobic bacterium, has been found in many silages [46,47,48]. It is a detrimental microorganism to silage, and its growth is usually inhibited in a low pH environment [49]. In this study, the abundance of *Klebsiella* in the enzyme groups gradually decreased as the ensiling time prolonged, which may be related to their lower pH values. *Lacticaseseibacillus* and *Lactiplantibacillus* are homofermentative lactic acid bacteria. *Lacticaseseibacillus* was the dominant lactic acid bacteria in the early stage of ensiling, and the dominant lactic acid bacteria changed to *Lactiplantibacillus* in the post-ensiling stage [50]. This indicated that the fermentation of the mixed silage of king grass and rice straw was dominated by homofermentation. Consistently, Liu et al. showed that the abundance of *Lactiplantibacillus* was positively correlated with WSC content [51], and *Lactiplantibacillus* had a positive effect on nutrient preservation [52].

According to previous studies, silage treated with complex enzymes (including cellulase and xylanase) can increase milk production in dairy cows, as well as improve digestibility and rumen fermentation ability in sheep [53,54]. This diet is of great significance for improving the nutrition of ruminants. Similarly, our research showed that the application of appropriate doses of cellulase and xylanase will have a positive impact on the mixed silage of king grass and rice straw, which may continue to benefit ruminants. In future research, we will conduct an in-depth study on the effects of mixed silage of king grass and rice straw on the nutritional requirements of ruminants and promote the development and utilization of silage.

## 5. Conclusions

The present study demonstrated that the inclusion of cellulase and xylanase in the mixed silage of king grass and rice straw resulted in an increase in LA and WSC content and a decrease in pH, NH_3_-N, and fiber component content. Moreover, the supplementation of cellulase and xylanase could enhance the abundance of beneficial bacteria such as *Lacticaseseibacillus*, *Lactiplantibacillus* and reduce the richness of the harmful bacteria *Klebsiella*. It is suggested that the addition of cellulase and xylanase would improve fermentation quality and nutrient preservation via reducing the content of fiber components, increasing the content of fermentation substrates and altering the bacterial community, with 1% cellulase or complex enzyme best.

## Figures and Tables

**Figure 1 microorganisms-12-00561-f001:**
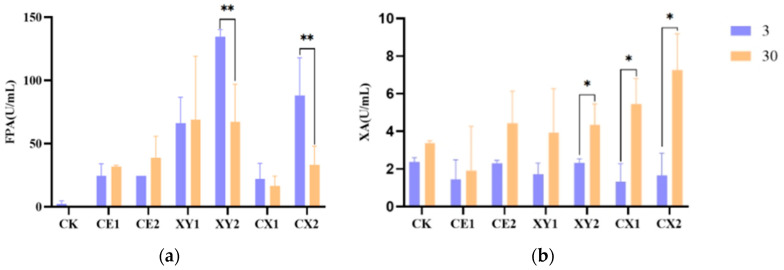
Enzyme activity of mixed silage of king grass and rice straw. (**a**) Filter paper enzyme activity on day 3 and 30 of ensiling fermentation; (**b**) xylanase activity on day 3 and 30 of ensiling fermentation; * *p* < 0.05; ** *p* < 0.01.

**Figure 2 microorganisms-12-00561-f002:**
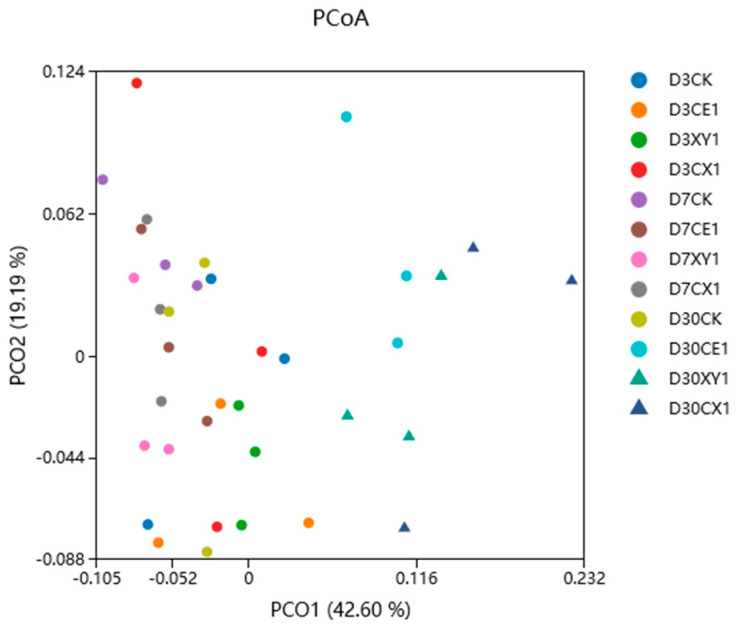
Principal coordinates analysis (PCoA, weighted UniFrac) of bacterial community in the mixed silage of king grass and rice straw. (D3CK: on day 3 of ensiling fermentation-blank control, silage ensiled without addition of additive; CE1, XY1 and CX1: silage ensiled with addition of 1% cellulase, 1% xylanase and 0.5% cellulase +0.5% xylanase on an FM basis).

**Figure 3 microorganisms-12-00561-f003:**
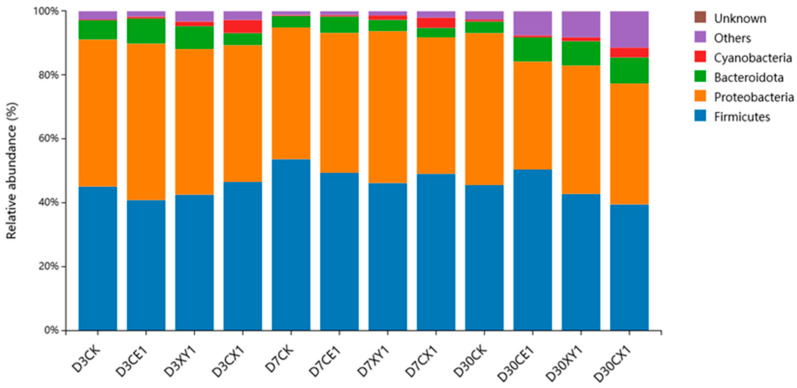
Bacterial community at the phylum level with relative abundance greater than 1% in mixed silage of king grass and rice straw. (D3CK: on day 3 of ensiling fermentation-blank control, silage ensiled without addition of additive; CE1, XY1 and CX1: silage ensiled with addition of 1% cellulase, 1% xylanase and 0.5% cellulase +0.5% xylanase on an FM basis).

**Figure 4 microorganisms-12-00561-f004:**
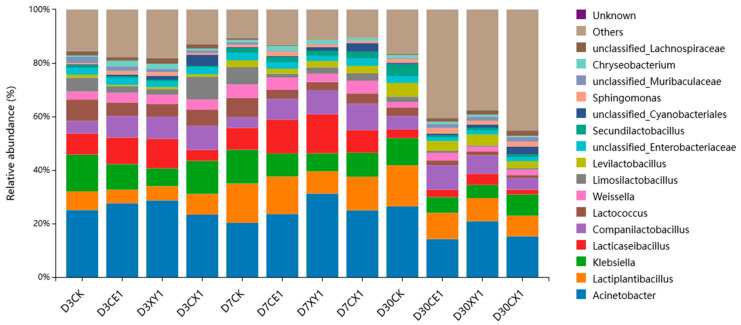
Bacterial community at the genus level with relative abundance greater than 1% in mixed silage of king grass and rice straw. (D3CK: on day 3 of ensiling fermentation-blank control, silage ensiled without addition of additive; CE1, XY1 and CX1: silage ensiled with addition of 1% cellulase, 1% xylanase and 0.5% cellulase +0.5% xylanase on an FM basis).

**Table 1 microorganisms-12-00561-t001:** Chemical composition of fresh king grass and rice straw used for silage production.

Item	King Grass	Rice Straw
Dry matter (% Fresh matter)	17.60	88.26
Crude protein (% Dry matter)	8.33	3.43
Neutral detergent fiber (% Dry matter)	70.69	72.30
Acid detergent fiber (% Dry matter)	35.99	39.33
Acid detergent lignin (% Dry matter)	4.62	5.08

**Table 2 microorganisms-12-00561-t002:** Fermentation parameter of mixed silage of king grass and rice straw.

Item	Treatment	Ensiling Days			SEM	*p*-Value		
		3	7	30		D	T	D*T
pH	CK	4.06 a	4.04 a	4.06 a	0.026	0.14	<0.01	0.16
	CE1	3.76 b	3.74 b	3.75 b				
	CE2	3.65 Bcd	3.72 Abc	3.68 Bbc				
	XY1	3.64 d	3.65 cd	3.65 bc				
	XY2	3.69 bcd	3.67 bcd	3.62 d				
	CX1	3.75 Abc	3.65 Bcd	3.70 Abc				
		3.72 Abcd	3.63 Bd	3.64 Bbc				
LA (% DM)	CK	3.88 b	4.58 b	5.08	1.208	0.30	<0.01	0.97
	CE1	7.22 ab	6.76 ab	6.57				
	CE2	7.46 a	6.28 ab	8.00				
	XY1	7.33 ab	8.82 a	7.62				
	XY2	8.41 a	8.36 ab	9.59				
	CX1	5.35 ab	6.96 ab	7.80				
	CX2	5.86 Bab	6.95 Abab	7.98 A				
AA (% DM)	CK	0.73 B	0.71 B	1.21 A	0.181	0.04	<0.01	0.98
	CE1	0.98	0.79	1.05				
	CE2	0.87	0.83	1.30				
	XY1	1.08	1.26	1.29				
	XY2	1.13	1.29	1.62				
	CX1	0.89	1.03	1.36				
	CX2	0.96 B	0.96 B	1.32 A				
NH_3_-N (% DM)	CK	0.32 Ba	0.32 Ba	0.88 Aa	0.036	<0.01	<0.01	0.03
	CE1	0.30 Bab	0.30 Bab	0.80 Aab				
	CE2	0.22 Bc	0.23 Bbc	0.55 Ac				
	XY1	0.22 Bc	0.22 Bc	0.70 Abc				
	XY2	0.23 Bbc	0.27 Babc	0.66 Abc				
	CX1	0.25 Babc	0.26 Babc	0.58 Ac				
	CX2	0.23 Bbc	0.20 Bc	0.61 Ac				

LA (% DM), lactic acid (% dry matter); AA (% DM), acetic acid (% dry matter); propionic acid and butyric acid was not detected; NH_3_-N (% DM), ammonia nitrogen (% dry matter); SEM, standard error of the means; D, the effect of ensiling days; T, the effect of enzyme additives; D*T, the interaction of ensiling days and enzyme additives. Means with different lowercase/uppercase letters in the same row/column differ at *p* < 0.05.

**Table 3 microorganisms-12-00561-t003:** Chemical composition of mixed silage of king grass and rice straw.

Item	Treatment	Ensiling Days			SEM	*p*-Value		
		3	7	30		D	T	D*T
DM (% FM)	CK	22.50	24.08	21.59	1.086	0.48	0.62	0.85
	CE1	23.59	22.30	22.94				
	CE2	23.36	23.19	21.75				
	XY1	21.35	23.05	21.11				
	XY2	22.18	20.98	21.36				
	CX1	23.02	21.85	22.93				
	CX2	23.15	22.23	22.65				
WSC (% DM)	CK	0.06 Cc	0.28 Bc	0.65 Ac	0.454	<0.01	<0.01	<0.01
	CE1	0.93 Bab	0.41 Bbc	2.35 Abc				
	CE2	1.23 Ba	0.48 Bab	3.17 Aab				
	XY1	0.38 Bbc	0.54 Bab	5.52 Aa				
	XY2	0.30 Bbc	0.61 Ba	4.88 Aa				
	CX1	0.46 Bbc	0.46 Babc	4.97 Aa				
	CX2	0.12 Bc	0.51 Bab	4.65 Aab				
NDF (% DM)	CK	68.26 Aba	70.14 Aa	66.89 Ba	1.050	<0.01	<0.01	0.17
	CE1	63.07 b	62.91 b	58.14 b				
	CE2	62.58 Ab	62.15 Abc	57.86 Bb				
	XY1	62.91 Ab	61.31 Abc	56.52 Bb				
	XY2	62.11 Ab	58.49 Bd	56.56 Bb				
	CX1	61.88 Ab	61.83 Abc	54.09 Bb				
	CX2	60.06 Ab	59.76 Acd	55.61 Bb				
ADF (% DM)	CK	35.29 Ba	38.21 Aa	36.15 Ba	0.715	<0.01	<0.01	0.04
	CE1	32.49 b	31.95 b	30.87 ab				
	CE2	32.77 b	32.19 b	31.61 b				
	XY1	32.48 Ab	32.17 Ab	28.97 Bab				
	XY2	30.85 b	29.22 c	29.18 ab				
	CX1	32.71 Ab	31.72 Ab	28.72 Bb				
	CX2	31.48 b	30.53 bc	29.34 ab				
ADL (% DM)	CK	4.07 B	5.57 Aab	4.15 B	0.276	<0.01	0.04	0.25
	CE1	4.71	4.22 d	4.03				
	CE2	4.82	5.63 ab	4.64				
	XY1	5.14 AB	5.89 Aa	4.45 B				
	XY2	4.77	4.94 bc	4.26				
	CX1	4.98	4.60 cd	4.45				
	CX2	4.66	4.85 cd	3.78				

DM (% FM), dry matter; FW, fresh matter; WSC, water-soluble carbohydrate; NDF, neutral detergent fiber; ADF, acid detergent fiber; ADL, acid detergent lignin; SEM, standard error of the means; D, the effect of ensiling days; T, the effect of enzyme additives; D*T, the interaction of ensiling days and enzyme additives. Means with different lowercase/uppercase letters in the same row/column differ at *p* < 0.05.

**Table 4 microorganisms-12-00561-t004:** Alpha diversity of bacterial diversity of mixed silage of king grass and rice straw.

Item	Treatment	Ensiling Days			SEM	*p*-Value		
		3	7	30		D	T	D*T
Shannon	CK	5.18	4.93	5.22 b	0.25	<0.01	<0.01	<0.01
	CE1	5.46 AB	5.07 C	6.81 Aa				
	XY1	5.54 B	4.86 C	7.14 Aa				
	CX1	4.97 B	4.83 B	7.52 Aa				
Simpson	CK	0.93	0.92	0.91 b	0.01	<0.01	<0.01	0.01
	CE1	0.94	0.93	0.96 a				
	XY1	0.95 B	0.93 C	0.97 aA				
	CX1	0.92 B	0.93 B	0.97 aA				
Chao1	CK	1144.47 C	1361.63 B	1753.86 A	75.49	0.12	<0.01	0.28
	CE1	1190.27 B	1406.58 B	1740.20 A				
	XY1	1366.88 B	1356.11 B	1776.59 A				
	CX1	1225.17 B	1459.28 B	2008.01 A				
Ace	CK	1396.75 B	1771.80 A	1738.45 A	105.01	<0.01	0.10	0.51
	CE1	1410.83	1748.33	1601.87				
	XY1	1637.24	1871.92	1602.43				
	CX1	1510.46	1966.78	1925.63				
Coverage	CK	0.99	0.99	0.99	-	-	-	-
	CE1	0.99	0.99	0.99				
	XY1	0.99	0.99	0.99				
	CX1	0.99	0.99	0.99				

SEM, standard error of the means; D, the effect of ensiling days; T, the effect of enzyme additives; D*T, the interaction of ensiling days and enzyme additives. Means with different lowercase/uppercase letters in the same row/column differ at *p* < 0.05.

## Data Availability

The datasets presented in this study can be found in online repositories. The names of the repository/repositories and accession number(s) can be found at https://www.ncbi.nlm.nih.gov/sra/ (accessed on 1 February 2024), PRJNA1070481.

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
