# Peer review of "Effects of Cellulase and Xylanase on Fermentation Characteristics, Chemical Composition and Bacterial Community of the Mixed Silage of King Grass and Rice Straw"

_microorganisms, 2024, doi:10.3390/microorganisms12030561_

Round 1
Reviewer 1 Report
Comments and Suggestions for Authors
Dear authors,
I have reviewed your manuscript. Below you will find some comments, which I hope will be useful.
Introduction:
Line 36: King grass belongs to the Pennisetum genus of the Poaceae or Gramineae family
Line 46-49: For a better understanding, it would be necessary to mention the moisture content of King grass as well as the moisture content suitable for ensiling. In fact, moisture content is one of the most important factors when ensiling, determining a proper fermentation process. In this way, it is easier to “connect” the idea of ​​the following paragraph.
Line 60-68: It is necessary to give some background on the use of additives for silage and describe what cellulases and xylanases are and their uses in livestock feeding.
Materials and methods
Line 88-90: It is not clear why three bags were not sealed. Doesn't it contradict what was mentioned previously?
Line 107-108: What do you mean by previous methods?
Line 110-111: Why was the bacterial community analysis not performed with 2%?
Results
Note: Tables should be placed after they are mentioned in the text. Please improve the style and format of the tables to make interpretation easier and clearer.
Line 134-140: The data reported in the tables should not be repeated. Instead, describe them in another way.
Line 149: What do LA and AA mean? If they are lactic acid and acetic acid, indicate this from the first time they are mentioned in the text.
Line 160: What do PA and BA mean? If they are propionic acid and butyric acid, indicate this from the first time they are mentioned in the text.
Line 163: Crude protein is not described. Why?
Line 198-200: indicate what * and ** mean in figure 1.
Discussion
Given the number of treatments, sampling times and the variables evaluated, I highly recommend further discussion of the results. There are many studies on the use of cellulases and xylanases in the diet of ruminants, including silages of different forages. Check the reference articles.
Comments on the Quality of English LanguageThe English language is understandable. Some minor to moderate changes are required.
Reviewer 2 Report
Comments and Suggestions for Authors
I evaluate the reviewed article very positively. The research is important because in most cases, the use of biomass is possible provided it is available throughout the year. For this reason, it is important to look for solutions that will make this possible. The purpose of the research was clearly stated. The introduction is interesting and explains the need to conduct research, the results of which are presented in the article. The methods used were presented in a manner that does not raise any objections. The results were described clearly. The only question that raises some doubts is the way in which the tables and figures are arranged, as I describe below. The discussion chapter, in which the authors explain their results, is unobjectionable.
Detailed notes:
1. Line 81 – why was this ratio adopted?
2. Line 145 – please try to place Table 2 closer to the reference in the text
3. Table 1 – please change the title of the table. These are not chemical properties.
4. Table 3 – please change the title. There is nothing in the table about nutrients. Please move the table closer to line 164.
5. Figure 1 – Please move closer to Line 178.
6. Figure 2 – please move closer to Line 214.
7. Figure 3 – no reference in the text.
8. The conclusions should be extended.
I believe that after taking into account these few comments, the article can be published in Microorganisms.
